# Educational Influencers on Instagram: Analysis of Educational Channels, Audiences, and Economic Performance

Javier Gil-Quintana [ID] and Emilio Vida de León *[ID]

Department of Didactics, School Organization and Special Didactics, National University of Distance Education, 28040 Madrid, Spain; jgilquintana@edu.uned.es
* Correspondence: emilio.tmz@gmail.com

**Abstract:** Influencers have positioned themselves as opinion leaders capable of influencing large social groups, extending their presence to areas such as education. Instagram is one of the most consolidated social networks focused on the image where citizens interested in educational areas can find information from specialized channels on this topic. The purpose of this study is to analyze, during the period of confinement by COVID-19, the use of Instagram by educational influencers to consolidate their channel in new audiences, influence through interaction with their followers and create their transmedia production. Using a mixed methodological approach, a descriptive analysis of a sample of 810,200 users and a content analysis of 13 profiles of educational influencers is applied. The results show educational influencers as true experts in the use of Instagram, managing visually pleasing and harmonious profiles for new audiences. These influencers reach a large number of users, mostly women between the ages of 25 and 45 with an interest in "motherhood" on the platform. Educational influencers use digital marketing codes in their social networks, with a communicative style adapted to this type of space that seeks to increase the interaction and participation of new audiences and, as a consequence, economic profitability. There is a high number of influencers whose objective is to share educational resources, using their accounts as showcases for their transmedia educational production and for the sale or promotion of their productions and creations.

**Keywords:** Instagram; educational influencers; communication; social media; new audiences; transmedia production; digital marketing; descriptive analysis; content analysis; social networks

## 1. Introduction

Social networks have been consolidated in society, during the COVID-19 pandemic period, as tools for interaction between people from different parts of the world. The study and analysis of the communication and interaction strategies applied by users in these environments are used from the digital marketing sectors to define strategies with which to reach wider audiences and adapt to the new ways in which society consumes information. In this aspect, the figure of influencers has established itself as one of the most important, thanks to its ability to set the trends of the moment, having a fundamental role in the consumption of a large part of the population. In addition to the perspective of digital marketing, in recent years the education sector has made the leap to these digital environments in which to expand its transmedia production, known, as collected by Jenkins and Scolari, as that flow and connection that occurs through multiple distribution channels and platforms [1,2], involving the different social agents (students, families, etc.). The purpose of this work is focused on analyzing how educational influencers use the social network "Instagram" to show their transmedia production to audiences. This study focuses on the analysis of 16 accounts of educational influencers in Spain (characterization of the channels and content analysis), the impact they generate on their audiences and the economic benefits they generate with their activity on Instagram. An exploratory and descriptive study has been applied for the approval of research propositions. One of the main values of this research is the contribution to the scientific community on the figure of

the educational influencers in our society, as well as clarifying what type of activity they carry out and the relevance of their reach on Instagram.

### 1.1. Instagram as a Social Platform

The post-COVID-19 society in which we live in 2021 is immersed in an uncertainty in which digital technologies govern people's daily lives, generating situations of real instability and dependence in their absence [3]. The communicative model that has been empowered through new media has seen its maximum projection in social networks such as Facebook, Instagram, Twitter, TikTok or Twitch; this fact reminds us once again that human beings are social beings who constantly seek interaction with their peers, thus turning social media into fundamental channels for instant communication and interaction between people in the global village [4]. This fact has disrupted the activity of traditional power (political parties, media and companies) [5], forcing them to make a considerable leap towards citizen participation environments in order to disseminate their productions and influence the social imaginary [6]. In the midst of this media framework, there is a danger of generating disinformation, in this case, fake news being the current hot topic, distancing citizens from the development of critical thinking in the context of so much media manipulation driven by ideological interests.

Whether to generate information or to provoke disinformation, nowadays there are extensive possibilities to interact in social networks, with digital access being quite common among citizens not affected by the digital gap. It is true that some social networks such as Facebook or Twitter have a large legion of users, but Instagram currently enjoys greater success among the younger generations as it is a very attractive social network that allows for production and creation in image, audiovisual and hypertext formats and stands out for being highly intuitive [7]. Instagram in its early days offered a more limited service than the current one, as in recent years the platform has added new functions in response to technological advances and user demands, including: Instagram Stories (24 h stories), the highlights section (space to keep stories on the profile), Instagram Shopping (tagging products in publications) and IGTV (option to upload vertical and long videos) [8]. For these reasons, Instagram is recognized as the leading image-centric social network, offering its users constantly updated content based on the accounts with which users interact, profiles followed, likes, etc., in which static images alternate with long or short videos. Despite its long life cycle, this social network is much consolidated, showing constant growth year after year [9]. The data provided by The Instagram Summary 2021 [10] shows that this social network is in a quite positive situation, showing a growth of 22.1% compared to 2020, reaching 1.221 million active users. Among the profiles created, there is a fairly even distribution of usage by gender, with 51% women and 49% men. In terms of Instagram age group use, the worst results are obtained among 13–17-year-olds (7.3%), 55–64-year-olds (3.8%) and +65-year-olds (2.1%). These data improve in groups between 35–44 years old (16%) and 45–54 years old (18.1%). The population groups that lead the results in terms of the use of this social network are those aged 18–24 (29.8%) and 25–34 (33%), accounting for some 766.7 million Instagram users. Digital natives, or millennials, young people born between the 1980s and 2000s, are the main consumers of social networks, since they perfectly understand the codes that are used in them [11–15]. As children of the Baby Boomer Generation, they have lived in the digital context, conditioned by other social values, and other ways of working and relating. Millennials seek interaction in these environments, feeling interested in the information they find on the Internet in order to update themselves, study, interact, consume and be entertained; it is a hyper-connected generation that still resists trends. This generation shows great social and ethical values, according to a report published by CIBBVA in 2021 [16]. Alsop defines this generation as tolerant, optimistic, restless, civic, team-oriented and conscientious people in the search for a balance between work and leisure [17]. Syrett and Lammiman describe them as individualistic, technological, sophisticated, mature and with a great personal identity [18]. In the social sphere in which Burstein has lived, the economic crisis in which this generation

has grown up stands out, causing an activist and protest profile, although affected by the consumer culture and materialism that has been projected with force through the networks social [19].

*1.2. Influencers*

Social networks have not only changed the form of participation and interaction of citizens and the creation of a culture of participation [20] but also the possibility of consuming information on the part of new audiences, leading to a change of role to a direct, immediate and purely participatory interaction, going from being passive consumers to being active, critical actors and creators of content as prosumers [21]. Interaction is part of the very nature of human beings in their search for belonging to a group, which in these digital environments translates into the implementation of certain elements such as likes, mentions, comments, hashtags, etc., which favour the loyalty and commitment of audiences and their engagement [22]. This type of interaction favours the growth of Instagram profiles, especially among those who generate the most activity among their audiences: influencers.

Influencers are people who have great power of influence in social networks, with a large number of followers and with great prestige among children, adolescents and young people. The key aspect of a good influencer and the one with the most presence of brands is their engagement power that they enhance with their target. The ability of a product or a brand to establish solid and lasting relationships with their followers is demonstrated, creating a commitment that is established between the brand and consumers and the consumption habits that it develops. The figure of the influencer can be defined as users who are experts in a subject matter and who have a certain power or prestige to make their creations and transmedia productions reach a large number of people, generating a notorious repercussion on new audiences, sometimes even influencing the tastes and trends of society. In this sense, we can understand the transmedia production as the story that each user shares on communication platforms, in this case Instagram, being recorded as their own personal mark, a trace that consumers are responsible for expanding on the network through their actions (likes, shares, etc.). Well-known influencers make constant use of strategies to control their audiences and traffic of channel consumption, to access followers with a certain profile, to organize the content offered on the channel, to manage and schedule publications or to manage the profile's links [23]. Around all these approaches, various theories have been developed that explain the basis for the growth of new audiences on social networks, and which are supported by the various algorithms. Peters et al. [24] argue that the elements that favour user growth on Instagram include the quality of the content and the quantity of posts made, with the former having greater weight. Sánchez-Amboage, Membiela-Pollán and Rodríguez-Vázquez [25] establish five keys for effective communication on Instagram in order to grow a brand: (1) Use the image as the main resource; (2) generate a feed with personal style that acts as a connecting link; (3) include text message in the image description, using a language close to the audience; (4) use humor as a key to entertainment; and (5) be constant in publishing content on the channel. The content generated by these influencers can promote an informed, committed and critical citizenry, although also on occasions, as we have observed due to the COVID-19 pandemic, they have jeopardized an adequate practical implementation of the basic health principles: autonomy, beneficence, justice and non-maleficence [26].

This context has generated a breeding ground for the development and implementation of so-called digital marketing [27], which has to do with the commercial use given to social networks by companies, media and political parties, and of course, also by individuals such as influencers, in order to market their products in the midst of platform capitalism. Digital marketing seeks to take advantage of interactivity and segmentation, immediacy, globalization, quality of impact, accessibility and reduced economic costs to position brands among social network users. Part of the strategies followed by many companies on the Internet are based on influencer marketing, which consists of the pro-

motion carried out by the influencer through their channel, with the aim of generating a positive repercussion among their audiences towards the promoted brand [28]. Social media marketing and the strategies associated with it, however, are not relevant to all generations, with millennials being the group most attuned to the type of advertising and online commerce, a fact known among marketers to establish appropriate strategies [29].

*1.3. Educational Influencers on Instagram*

Social networks have become essential spaces for interaction and entertainment also in professional life [30], including areas of human development such as education. However, all the areas of interest on Instagram share the same characteristic: They include influencers who use their status on the network to influence audiences [31]. As it can be intuited, the educational world, in its constant search for innovation and adaptation to the digital reality, has sought its own ways to adapt to these new environments in which to share and give relevance to pedagogical aspects that may be of interest to the various educational agents. In this context, educational influencers emerge, who project themselves as leaders of the new audiences and who generate and share transmedia productions and creations related to this field [32]. Within the Instagram social network, different profiles of educational influencers are projected, as they are distinguished by different patterns depending on the type of content they offer on their channels. Studigrammers are characterized as users who share notes, doubts and educational reflections with their audiences [33]; bookstagrammers share content related to children's books or stories with the community, encouraging the desire to read [34]; learning influencers are teachers who use social networks such as Instagram to promote learning situations among their students or audiences, from the perspective of inter-creation and participation [35]. Likewise, in these environments, there are profiles of influencers who share didactic resources with the community, so that they can be applied in different pedagogical contexts [36], becoming a kind of learning mediators. It should not be forgotten, due to the great repercussions they tend to generate, the influencers who, within the educational framework, offer humorous content through memes [37], a format that is widely consumed among the younger generations. Finally, digital marketing has taken over a large number of the publications disseminated on Instagram; educational influencers are part of this mercantilist trend in the media, using their profiles as showcases for companies that request their services or to advertise and sell their own services to their audiences [38].

## 2. Materials and Methods

The research on social networks, in this case, Instagram, is really relevant due to the large amount of information it offers on large population constructs, providing data related to tastes, self-management or social interaction [39]. The study process addresses how educational influencers use Instagram to consolidate their channel with new audiences, influence the community and make transmedia projections of their productions and creations. In order to achieve this aim, it is appropriate to set objectives that serve as a reference point for the study:

- O1: Study the main characteristics of the profiles of educational influencers;
- O2: Discover the media impact that educational influencers have on new audiences;
- O3: To analyze the communication and interaction of educational influencers with new audiences through their transmedia production.

Likewise, the propositions established for this analysis are:

- P1: The audiences where educational influencers generate the greatest media impact are mostly millennials.
- P2: Educational influencers use a transmedia projection of the resource materials they produce and create for purely educational and non-commercial purposes.
- P3: Educational influencers establish effective channels of communication and interaction with their audiences.

This study is based on the approaches of mixed analysis models, in which quantitative and qualitative method approaches interact, with the aim of establishing the state of the question in depth [40]. In order to move forward in a rigorous and scientific manner, two categories are established, favoring a better treatment of data. The two categories are (a) educational influencers and new audiences; (b) transmedia production. In addition, it should be specified that there are several subcategories that are conducive to a more comprehensive study, knowing that educational influencers and new audiences are composed of: characterization of educational influencers, new audiences and media impact: communication and interaction. The second category, Transmedia production, is divided into: Transmedia image production and creation, Transmedia hypertext production and creation and Economic purpose of transmedia production (Table 1).

**Table 1.** Categories and subcategories.

| Categories | Subcategories |
| --- | --- |
| Educational influencers and new audiences | Characterization of educational influencers |
| | New audiences |
| | Media impact: communication and interaction |
| Transmedia production | Transmedia image production and creation |
| | Transmedia hypertext production and creation |
| | Economic purpose of transmedia production |

The quantitative paradigm allows us to advance in the study of the data obtained through the influencer tracking tools Heepsy and Influencity, as well as certain quantitative data (account type, main image, language of the channel, links on the board, design of the publications, characteristics of the message and codes used) obtained through a check list of the accounts analyzed on the Instagram platform. The use of the SPSS (IBM, Amonk, NY, USA) program in its Windows version has favoured the efficient and orderly processing of the data, making progress in the application of descriptive analyses on some dimensions of our research: characterization of educational influencers and impact on new audiences. The analysis of averages and percentages as descriptive analysis techniques has been applied to the following variables of the two blocks mentioned: educational influencers' characteristics (type of account, main image, language used and links on the board), follower profile (gender, age range and interests) and media impact (number of followers, number of likes, number of comments, number of posts, number of reactions, number of posts per day, comments per post and economic impact).

On the other hand, the qualitative approach has been used to analyze the creation and production of transmedia on the accounts of educational influencers as part of the last category of our research, content analysis. Taking into account the main characteristics of Instagram, we first decided to carry out an analysis of the visual content of the Instagram accounts, and secondly, we analyzed the discourse of these influencers in the written messages they provide as descriptions in their posts. As part of the analysis of the visual content on Instagram, we studied the esthetics of the publications, paying attention to the visual feed offered by the boards of the educational influencers and, secondly, the use of personal images as part of the content. Within the discourse analysis, elements such as the type of hypertext, use of formal or informal language, incitement to audience participation, types of interaction elements used (hashtags, mentions, emoticons, links, etc.) have been analyzed. Finally, as part of the content analysis, we proceeded to analyze the purpose of the publications, systematically recording the purpose for which educational influencers use their accounts: commercial, materials, training, memes, personal, art, etc., and finally to be able to determine what type of influencers we are dealing with.

*2.1. Investigation Process*

The process of this research is divided into four phases:

1. Data collection period. This phase took place between the months of March and July 2020. In the first place, a list of educational Instagrammers proposed by the Ministry of Education, Research, Culture and Sports of the Generalitat Valenciana [41] was used, thanks to which a total of 13 Instagram accounts of educational influencers were selected. Using "Heepsy" (Berango, Vizcaya, Spain) and "Influencity" (Madrid, Spain) software helped the data collection. The use of direct observation of the accounts and the control lists made it possible to analyze the characteristics of the channels involved in our research.

2. Categorization. Once a first careful reading of the themes found in our sample had been carried out, it was considered important to define some categories of analysis that would contribute to a meaningful and coherent organization of the analysis. Two main categories were established with their corresponding subcategories (Table 1).

3. Analysis of the categories. The previous categories were analyzed through descriptive analysis (means and percentages) where the SPSS (IBM, Amonk, NY, USA) program helped us organize and present the information in an adequate way. Likewise, the content analysis of the analyzed accounts was very relevant where the use of checklists stood out to record each element studied.

4. Study of the results obtained. In this last phase, with the information organized into categories, the results obtained were described through an exhaustive study of de data collection.

*2.2. Sample*

Following the list offered by Eines Digitals Educatives on its website [28], we find the Instagram accounts with educational content that have been chosen to make up our sample of Instagrammers. The channels involved in this study offer content related to the educational area, sharing educational resources, humorous content or information for future teachers.

The sample included 13 Instagrammers and their 810,200 followers. Table 2 shows the accounts involved in this analysis, the number of followers they have and the first publication date on each channel. The data used in this study has been collected from March to July 2020, coinciding with the closure of the schools as result of the outbreak caused by the COVID-19 pandemic.

**Table 2.** Research sample.

| Account Name | Followers | First Publication Date |
| --- | --- | --- |
| @2profesenapuros | 63,000 | 8 March 2015 |
| @3ways2teach | 6800 | 29 September 2015 |
| @applesandabcs | 150,000 | 23 October 2012 |
| @auladeapoyo | 41,000 | 22 August 2016 |
| @clubpequeñoslectores | 37,000 | 17 October 2014 |
| @desdemiaula | 13,000 | 21 December 2016 |
| @educacioilestic | 14,000 | 10 September 2015 |
| @enticonfio | 7400 | 6 February 2014 |
| @entrenubesespeciales | 120,000 | 6 May 2016 |
| @maestradepueblo | 130,000 | 30 January 2016 |
| @maestrosaudicionyl | 31,000 | 3 January 2016 |
| @teachinghumor | 67,000 | 8 January 2014 |
| @thinksforkids | 130,000 | 9 August 2013 |

## 3. Results and Findings

The descriptive analysis applied in this study allows us to establish a clear framework for the state of the characterization of educational influencers, new audiences and the media

impact that these influencers generate on their channels, thanks to the results obtained through a statistical analysis of means and percentages applied to the different analysis selected for this research.

The analysis of the visual content and the discourse in these channels offers an identification of the analyzed accounts and the processes followed to reach their audiences. In this sense, we include the Transmedia production category and its subcategories. A descriptive analysis is added on the average earning data of the accounts involved.

### 3.1. Educational Influencers and New Audiences

3.1.1. Characterization of Educational Influencers

A large majority of the influencers analyzed (85%), as can be seen in Figure 1, use an account that does not have verification, which means that they are not officially recognized as a public image and may be more exposed to account falsifications; the following profiles do have such verification: @maestradepueblo and @thinksforkids. The use of the main image of the account on Instagram is the introduction letter of any influencer on social networks. It can be seen that the majority of educational influencers (69%) have decided to use an identity image in relation to the content that can be found on the channel; however, @applesandabcs, @maestradepueblo, @maestrosaudicionyl and @thinksforkids use their personal image as the main avatar of the account. The language most used in the accounts analyzed is Spanish (62%), followed by English (15%), with some accounts using several languages to have a greater impact on new audiences (Spanish, Catalan and English). Within the account description, 92% of influencers have added a link to a website external to Instagram, and 46% of the accounts analyzed make use of a link manager (linktree, linkbio or linkinprofile) in which the audience accesses a list of web spaces to which they can go via transmedia navigation, proposed by the influencer in question (personal page, personal blog, shopping page, others, etc.). Moreover, 38% of the profiles in this study link directly to their personal website, where they share information about personal productions, materials they have created, training services or sales of other products. Finally, we have only found one account that also screens on the YouTube social network with its audiences and another that has no link enabled. As we can see in Figure 1, a diagram shows the main characteristics that these influencers offer to their audiences in their home profile, compiling information related to the type of account, the main image, the language used or the use of external links. In 100% of the biographies of these profiles, written text has been used in which explicit and concise information is given about the type of content that can be found in the profile, but not about their personal background, academic training or qualifications, thus generating a disinformation bias.

In the analysis of the new Instagram audiences, it is interesting to know, as part of the characterization, the credibility of the users who make up the followers of each of the educational influencers analyzed; Heppsy software has given us some interesting facts about it. Figure 2 shows the percentages of nice followers, which refers to real followers who carry out their activity without the presence of computer programs (Bots). Likewise, the doubtful followers appear on this figure representing those accounts thar present suspicious activity typical of Bots.

In all cases, the percentage of 'nice followers' is above 80%, which means that all influencers have a quality audience. It can be seen that the account with the worst data (@thinksforkids = 83.79% 'nice followers') and the one with the best percentages (@maestradepueblo = 92.03% 'nice followers') are also two of the accounts with the highest number of followers, which is interesting as it may have an impact on interaction levels. To determine the credibility scores of followers, factors such as avatar, biography description, number of posts or number of accounts followed vs. follower ratio are taken into account. Influencers with quality audiences will score 80 or more on the 'nice followers' variable.

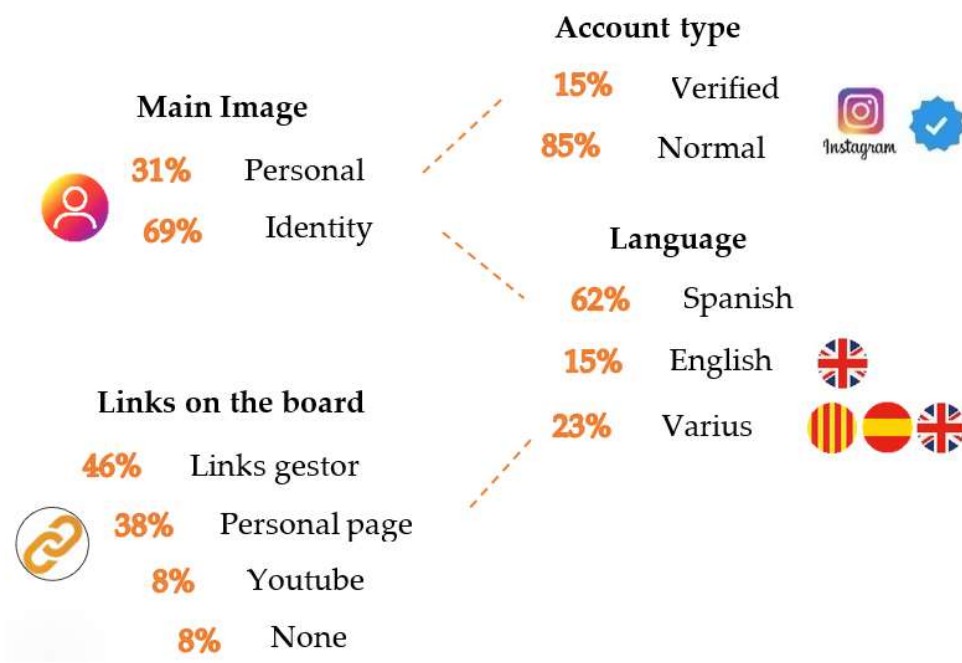

**Figure 1.** Characterization of the accounts. Source of own elaboration.

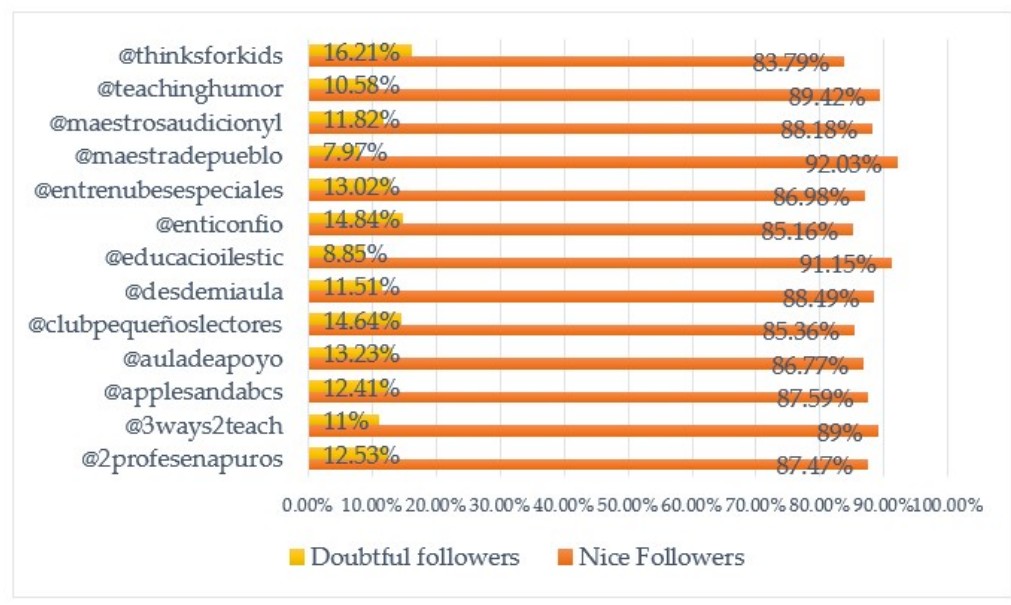

**Figure 2.** Percentage statistics (Quality of the audience). Source: Own elaboration.

### 3.1.2. New Audiences

In addition to the initial characteristics generated around influencers, it is appropriate to discover the new audiences that have been generated in the field of education observed since a demographic audience's characteristics point of view. It is interesting to note that of the more than 810,200 users who follow these accounts, 67% of the users (542,834) are female, while the remaining 33% are male (267,366), as shown in Figure 3. This data fluctuates slightly depending on the account we analyze, seeing how, for example, @teachinghumor has the largest female audience, with 73% of the total, while the @enticonfio account is the one with the highest percentage of men, with 43% of the audience being male.

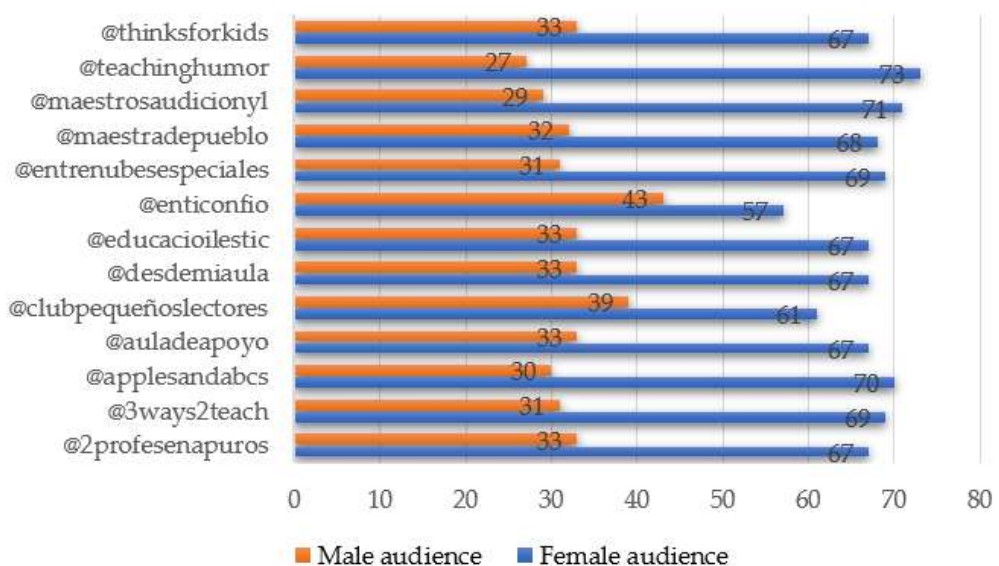

**Figure 3.** Percentages by gender. Source: own elaboration.

The age of the followers is another element that generates great interest when analyzing the characteristics of these audiences. For this purpose, we have analyzed the data obtained from the Heepsy tool (Berango, Vizcaya, Spain), which presents the audience of each influencer according to age range (–13 years, 13–18 years, 18–25 years, 25–35 years and 35–45 years), as shown in Figure 4. It can be seen that the age range with lower percentages is from the 13 years old to younger range (1.69%). This is closely followed by the audiences aged 13–18 (2.07%) and 18–25 (7.23%). Followers between 35–45 years of age have the best figures, rising to 26.08% of the audience. The age group that is shown to be the biggest consumer of this type of content, with a considerable difference from the rest, is the 25–35 age group, with a percentage slightly above 60% of the total audience, which is quite significant as it represents a total of 490,495 followers.

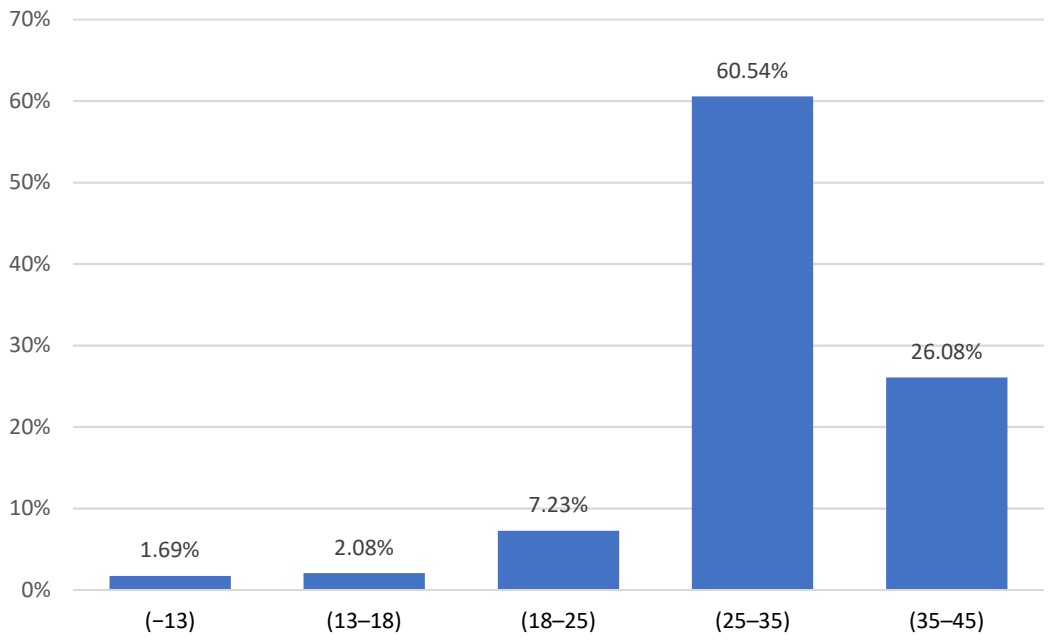

**Figure 4.** Percentage statistics of the means (Age). Source: own elaboration.

As for the interests of the audiences, as shown in Table 3, it can be said that 25.53% of the followers involved in this analysis show interests related to the category of motherhood,

representing around 207,000 thousand users, a figure that may be related to the age of the followers analyzed in the previous point. Other outstanding interests among the followers of educational influencers are: Pets 14.09%, Healthy Living 8.23% (fitness, sports and food) and Audiovisual Media 8.16% (TV, photography or graphic design). Below, with lower rates, are interests such as: travel 4.9%, beauty 4.08% or DIY 1.92% (Do It Yourself).

**Table 3.** Descriptive statistics (Audience interest).

| | N | | Media | Desv. Desviation | Minimum | Maximum |
| | Valid | Lost | | | | |
| --- | --- | --- | --- | --- | --- | --- |
| Maternity | 13 | 0 | 25.5385 | 11.93465 | 0.00 | 43.00 |
| Pets | 13 | 0 | 14.0923 | 9.43102 | 0.00 | 35.00 |
| Travels | 13 | 0 | 4.9615 | 4.71709 | 0.00 | 13.00 |
| Healthy life style | 13 | 0 | 8.2308 | 5.99171 | 0.00 | 24.00 |
| Bloggers | 13 | 0 | 2.4615 | 3.30695 | 0.00 | 8.00 |
| Interpretation | 13 | 0 | 2.3077 | 5.02302 | 0.00 | 18.00 |
| Beauty | 13 | 0 | 4.0846 | 3.84184 | 0.00 | 12.00 |
| DiY and Crafts | 13 | 0 | 1.9231 | 3.79608 | 0.00 | 11.00 |
| Music | 13 | 0 | 2.1154 | 3.04243 | 0.00 | 9.00 |
| M. Audiovisual | 13 | 0 | 5.0154 | 8.16954 | 0.00 | 29.00 |
| Youtubers | 13 | 0 | 0.4615 | 1.66410 | 0.00 | 6.00 |

### 3.1.3. Media Impact: Communication and Interaction

In terms of media impact, it is interesting to analyze the results obtained, as they offer a clear perspective on the characteristics of the communication model, the movement and interaction that takes place among the audiences of the accounts analyzed, helping to determine values regarding the impact that educational influencers generate on Instagram through the publication of content on their channels. Table 4 shows interesting data on audience ratings, the number of posts made, along with the engagement rate. We see that, as of 31 July 2020, the educational influencers with the highest number of followers are @applesandabcs, @thinksforkids, @maestradepueblo and @entrenubesespeciales, all of them with more than 100,000 followers. With regard to the Followers/Followings ratio, it can be said that there are no significant differences between the accounts with the most followers and those with the fewest followers, with an F/F ratio of around 50; however, we could highlight the high values regarding this ratio by @teachinghumor (F/F ratio = 600), @maestradepueblo (F/F ratio = 460) or @educacioilestic (F/F ratio = 140). The three accounts with the highest number of followers: @applesandabcs (150,000), @thinksforkids (130,000) and @maestradepueblo (130,000) are among the accounts with the highest number of posts, with 2700, 4500 and 1100. It is worth highlighting the high levels of transmedia content production by @2profesenapuros (2100 posts) and @enticonfio (1000 posts), which are at the same level in terms of number of publications as the accounts with the highest number of followers. Likewise, when analyzing the three educational influencers with the highest number of weekly publications (@2profesenapuros, @applesandabcs and @entrenubesespeciales), we can see that they are also the ones that show the highest growth rate, so it could be said that the higher the number of publications, the higher the growth rate of the channel, although this does not occur proportionally.

As part of the analysis of media impact, it is of particular relevance to study the scope of repercussions generated by the publications of the selected influencers. The data obtained show, firstly, that many more likes than comments are generated among the audience, both in photo publications and in those publications in audiovisual format, as shown in Table 5. Although @applesandabcs is the account with the highest number of followers and one of the accounts that produces the most content throughout the week (5.9 posts per week), it is not the one that reports the highest number of likes and comments. This contrasts with educational influencers @maestradepueblo (+130,000 followers) and @thinksforkids (+130,000 followers) who report high likes on image posts, with 3800 and

1600 likes, respectively. We should highlight the account @teachinghumor (67,000 followers) for the high rate of participation it generates among its audience through 'likes', 2600 likes on photo posts and 1000 likes on video posts. As for the comments generated by these influencers' image posts, they remain fairly low, at around 50 comments for those accounts with the most interaction. However, it is interesting to note that the @thinksforkids account generated more comments than likes on its image posts, producing a different communication to the rest of the accounts: 2900 comments for 1600 likes.

In terms of audiovisual productions, not all accounts offer this type of content. In this case, it can be confirmed that the accounts that obtain the most reproductions of audiovisual publications are those with the highest number of followers (@applesandabcs, @maestradepueblo and @teachinghumor), although, once again, it is worth highlighting the account @2profesenapuros, which offers very high reproduction data considering that it has less than half the number of followers of those mentioned above, which speaks of a very active audience. In terms of comments, there is again a low rate of participation, around 50 comments, considering the number of reproductions and likes that these publications generate. In fact, if one looks at the ratio of comments per like, it can be said that there is no direct relationship between a high number of followers and a high ratio of comments per like (Table 5).

**Table 4.** Interaction of educational influencers.

| Influencer | Follower | Following | F/F Ratio | Post | Post/S | Growth Ratio | Engagment Rate | Punt. |
|---|---|---|---|---|---|---|---|---|
| @2profesenapuros | 63,000 | 1900 | 33 | 2100 | 7.7 | 2.9 | 0.6% | 51 |
| @3ways2teach | 6800 | 1400 | 4.7 | 280 | 1.4 | 0.7 | 1.2% | 61 |
| @applesandabcs | 150,000 | 3600 | 40 | 2700 | 5.9 | 2.6 | 0.4% | 52 |
| @auladeapoyo | 41,000 | 1100 | 37 | 940 | 1.9 | 0.2 | 0.4% | 52 |
| @clubpequeñoslectores | 37,000 | 940 | 39 | 970 | 0.79 | 0.5 | 1% | 53 |
| @desdemiaula | 13,000 | 1300 | 10 | 320 | 1 | 0.2 | 0.7% | 53 |
| @educacioilestic | 14,000 | 96 | 140 | 510 | 0.21 | 0.1 | 4.1% | 58 |
| @enticonfio | 7400 | 540 | 14 | 1000 | 28 | 1.8 | 0.2% | 56 |
| @entrenubesespeciales | 120,000 | 4500 | 26 | 970 | 5.1 | 5.6 | 0.7% | 54 |
| @maestradepueblo | 130,000 | 290 | 460 | 1100 | 2.4 | 1 | 2.9% | 55 |
| @maestrosaudicionyl | 31,000 | 410 | 75 | 830 | 3 | 1.3 | 0.7% | 54 |
| @teachinghumor | 67,000 | 110 | 600 | 280 | 0.9 | −0.5 | 4% | 57 |
| @thinksforkids | 130,000 | 1500 | 86 | 4500 | 3.1 | 1.4 | 3.5% | 59 |

**Table 5.** Impact of publications.

| Influencer | Likes Pictures | Comment. Pictures | Video Playback | Likes Videos | Comment. Videos | C/L Ratio |
|---|---|---|---|---|---|---|
| @2profesenapuros | 360 [a] | 10 [b] | 17,000 | 1300 | 86 | 2.8 |
| @3ways2teach | 77 [a] | 5 [b] | 730 | 57 | 10 | 6.5 |
| @applesandabcs | 540 [a] | 21 [b] | 11,000 | 430 | 44 | 3.9 |
| @auladeapoyo | 170 [a] | 9 [b] | - | - | - | 5.3 |
| @clubpequeñoslectores | 340 [a] | 14 [b] | - | - | - | 4.2 |
| @desdemiaula | 81 [a] | 4 [b] | 1900 | 110 | 29 | 4.9 |
| @educacioilestic | 550 [a] | 4 [b] | - | - | - | 0.7 |
| @enticonfio | 13 [a] | 1 [b] | - | - | - | 7.7 |
| @entrenubesespeciales | 720 [a] | 56 [b] | - | - | - | 7.8 |
| @maestradepueblo | 3800 [a] | 56 [b] | 26,000 | 2300 | 110 | 1.5 |
| @maestrosaudicionyl | 210 [a] | 15 [b] | 4900 | 220 | 4 | 7.1 |
| @teachinghumor | 2600 [a] | 49 [b] | 17,000 | 1000 | 40 | 1.9 |
| @thinksforkids | 1600 [a] | 2900 [b] | 7600 | 700 | 11 | 175.3 |

[a.] Total number of Likes on the last 12 photos posted. [b.] Total number of comments on the last 12 photos posted.

### 3.2. Transmedia Production

Content analysis reveals key information to discover the common practices of influencers in transmedia production on Instagram. An analysis of the accounts involved in our study has been carried out through a checklist. The data obtained provide us with information regarding the design of the publications or the characteristics of the messages and codes used, showed in Figure 5. A first approach is achieved by analyzing fundamental

elements of this platform, such as: the aesthetics of the publications, the characteristics of the productions and the codes used. After the systematic review of these accounts, we have observed main features in the media productions of the influencers that have been used in the development of this analysis, as presented in Figure 5.

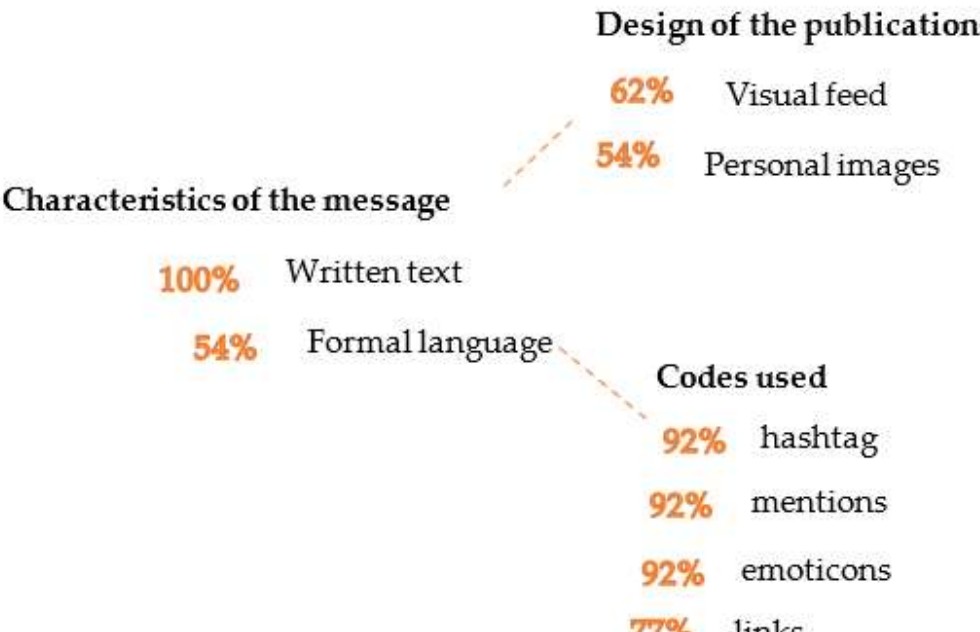

**Figure 5.** Analysis of publications. Source: own elaboration.

Jointly analyzing the transmedia production of the Instagram accounts involved in this study, allows us to offer a clear vision of the state of the research issue, complementing the data obtained with clear examples of the activities carried out by these influencers in their accounts.

3.2.1. Transmedia Image Production and Creation

The analysis of the images produced or created reveals that 62% of the influencers analyzed have a clear intention to take care of the aesthetics of the images that form part of their feeds, taking care of the quality projected from digital marketing (Figure 5). Some accounts, such as @entrenubesespeciales or @maestradepueblo, make use of a feed based on images with the same type of colour tones and filters, or using standardized frames and letters as part of the composition of the images posted. These types of elements are used as differentiating aspects with which to generate an image of the channel itself, being easily recognizable and pleasant for the audience, as can be seen in Figure 6. The image shows the main board of @entrenubesespeciales, composed of images with clear, clear backgrounds and related to educational resources.

The data on the publication of personal images as part of the content of the accounts is quite divided among the influencers under study. More than half of these accounts (54%) share personal images of the owner, selling his personal life, family members, work environment, etc. Although it should be noted that the production and creation of transmedia with personal images does not tend to be repeated in the media projection of their profile, but has more to do with specific situations in which influencers want to share with their followers highlights of their personal life with a clearly emotional and "hook" purpose, as can be seen in Figure 7.

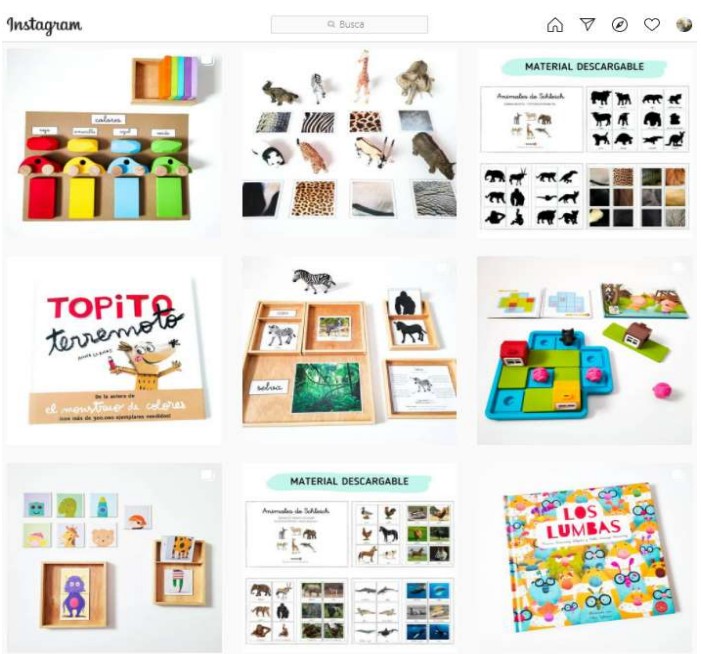

**Figure 6.** Example of an esthetic feed. Source: @entrenubesespeciales.

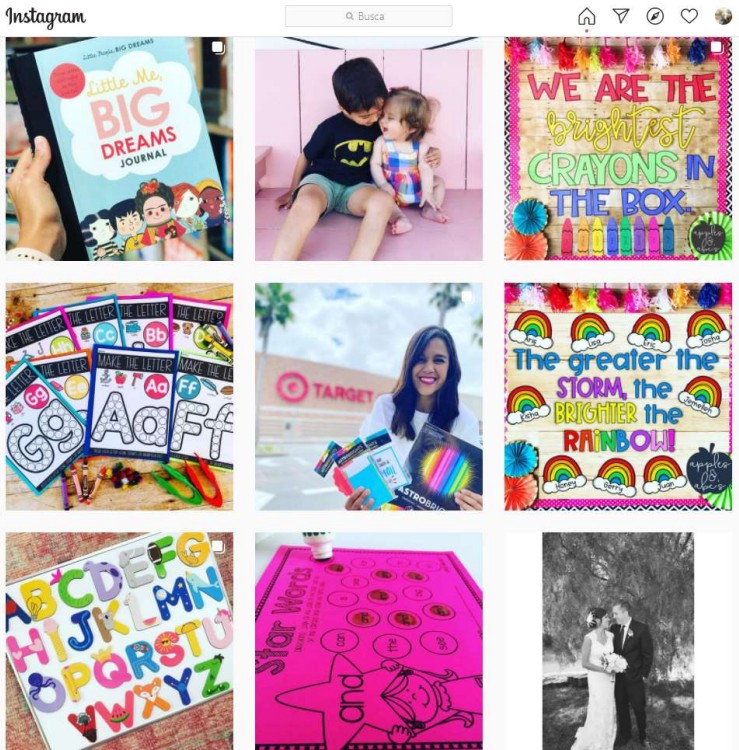

**Figure 7.** Example of the feed with personal images (@applesandabcs). Source: @applesandabcs.

### 3.2.2. Hypertextual Transmedia Production and Creation

Analyzing the transmedia hypertext production and creation of the influencers under study, it can be seen that 100% make use of hypertext to support the content of the images uploaded, providing the publication with more information and interest for the audience that consumes it (Figure 5). These hypertexts accompanying posts are often directly related to the image shared. In many of the cases observed, images of resource materials can be found as well as, in their description, a hypertext indicating the steps to create or access

them. Likewise, there is a division in the type of language used (formal and informal), with values very close to 50% in both types of language, between the language used in the educational area and the language used in influencer marketing. There is a tendency for educational influencers to produce and create hypertexts that incite audience participation and interaction, through various formulas: "What do you think?" "Do you know this game?" (@entrenubesespeciales); "Remember to upload it to the wall with the hashtag #talleresthinksforkids #thinksforkids and tag me so I can share it in stories and make a WORLDWIDE EXHIBITION" (@thinksforkids); Would you like me to do a FREE webinar teaching you how to make 4 board games in Canva? Leave me a comment on the post with your opinion" (@maestrosaudicionyl); as we can see in Figure 8.

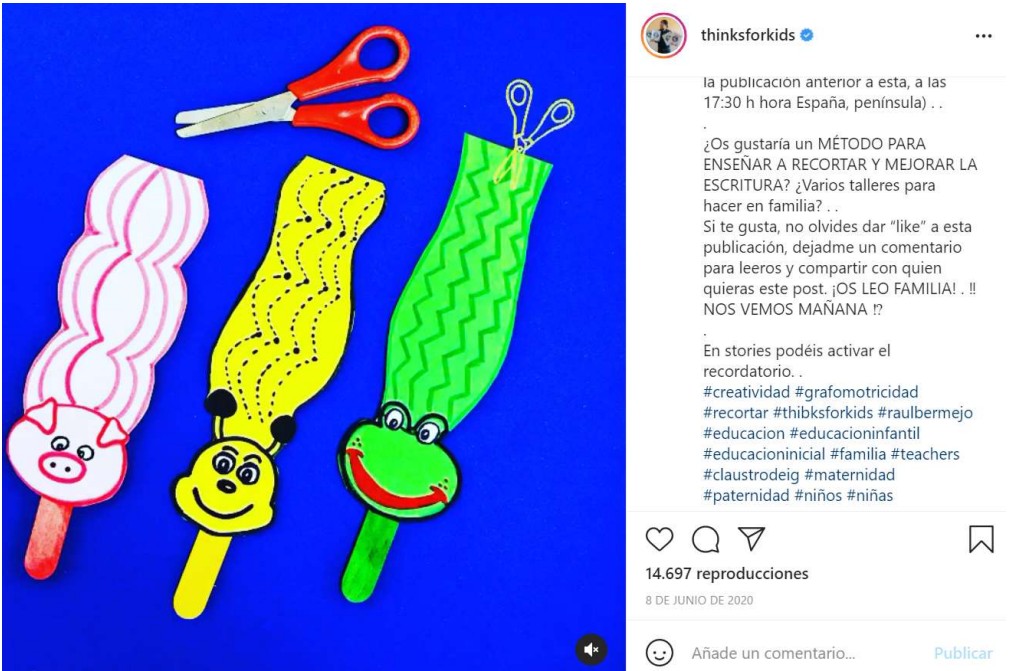

**Figure 8.** Example of the use of text in publications. In this case @thinksforkids interacts with his followers, asking about the content of the channel, and asking for likes on the publication Source: @thinksforkids.

Hypertextual transmedia production and creation projected on social networks have developed new textual codes to generate impact, acting as mechanisms of communication and interaction between users. It is interesting to observe how influencers make use of these codes in their publications to increase the transmedia projection of the published content. Throughout the hypertexts accompanying the productions and creations, we can see the high percentages regarding the use of hashtags, the introduction of mentions of other Instagram accounts and the use of emoticons. These percentages drop when it comes to inserting links in published texts or inviting people to visit other external Instagram pages (77%) (Figure 5). In general, these redirects encourage hyperlinked navigation to the influencer's own blogs or websites where they offer more information about other sales or services they offer to their audience; in this sense, we have observed that 92% of the accounts analyzed have a link to an external Instagram website in the header of their channel.

### 3.2.3. Economic Purpose of Transmedia Production

Table 6 shows what kind of transmedia production the accounts involved in our research share through their Instagram profile, using an analysis of the textual content of the publications made on their channels, interpreting the type of information that was shared in them between the months of March and July 2020. From the information collected

through a checklist, a categorization was created including the following objectives of publication: material, training, commercial, memes, private life, blog, art, formation, books or educational support.

When analyzing the purpose of the transmedia production that the influencers under study share on their accounts, it can be seen that in most cases they share ideas for material resources so that they can be used in learning contexts by their audiences (@2profesenapuros, @3ways2teach, @applesandabcs, @auladeapoyo, @desdemiaula, @entrenubesespeciales, @maestrosaudiciónyl and @thinksforkids), in some cases introducing a commercial purpose.

**Table 6.** Influencers' objectives according to the content of their profile.

| Influencer | Objective of the Publications |
|---|---|
| @2profesenapuros | Material/Training/Commercial |
| @3ways2teach | Material/Memes/Educational Support |
| @applesandabcs | Material/Private Life |
| @auladeapoyo | Material |
| @clubpequeñoslectores | Books/Commercial/Private Life |
| @desdemiaula | Material/Blog |
| @educacioilestic | Art |
| @enticonfio | Formation |
| @entrenubesespeciales | Material/Books/Commercial |
| @maestradepueblo | Memes/Commercial |
| @maestrosaudicionyl | Material/Formation/Commercial |
| @teachinghumor | Memes/Commercial |
| @thinksforkids | Material/Educational support/Formation/Commercial |

However, analyzing the hypertexts that accompany the production and creation of images, it can be said that some influencers such as @2profesenapuros and @maestrosaudicionyl have a clear intention to generate economic benefits from such materials by selling them on personal web pages, as presented in Figure 9. Other influencers, such as @clubpequeñoslectores or @entrenubesespeciales do not seek to make a direct financial profit by sharing these materials on Instagram, although they do share sweepstakes or discounts on materials and products from other accounts or companies.

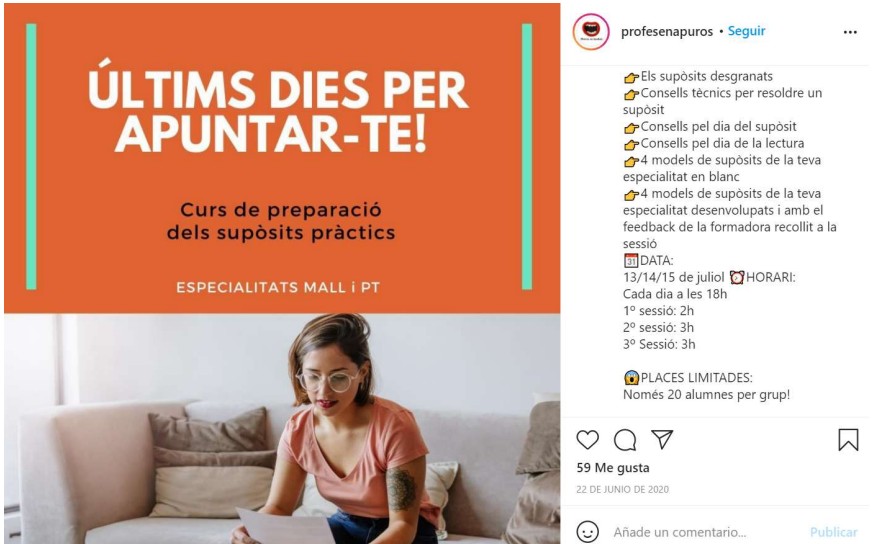

**Figure 9.** Example of publication for economic purposes. In the image @2profesenapuros shares different training programs and materials that are offered to its clients, limited by study places. Source: @2profesenapuros.

In addition to presenting educational materials, some of these accounts offer various educational training possibilities to their users. In the case of @2profesenapuros, they have a business structure through which they offer services for the preparation of competitive teaching exams, using Instagram as a channel for promoting these services. The @enticconfio account is part of a project promoted by the Colombian government to raise awareness and train education professionals in the responsible use of digital technologies at different educational levels, publishing content that gives visibility to the progress of its project and raises awareness of this issue among its audience, as shown in Figure 10.

Outside the perspective of practical approaches to the world of education, other accounts offer transmedia production focused on entertaining users, with themes related to education and children. Photographic art and aesthetic care are two aspects that are closely linked to social networks such as Instagram, following this line we can highlight the account @educacioilestic, whose account is clearly oriented to publications of artistic photographs related to childhood. Finally, other influencers (@teachinghumor and @maestradepueblo) can be observed using their accounts as spaces to share humorous publications about the world of education, using a format that is quite widespread in social networks, such as memes similar to the one shown in Figure 11.

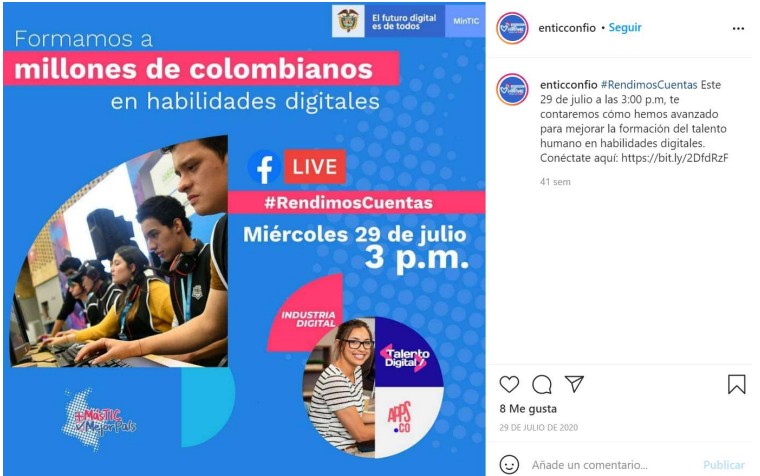

**Figure 10.** Example of a publication for educational purposes. This image shows the channel @enticconfio sharing with the community the date and online site for a free training session. Source: @enticconfio.

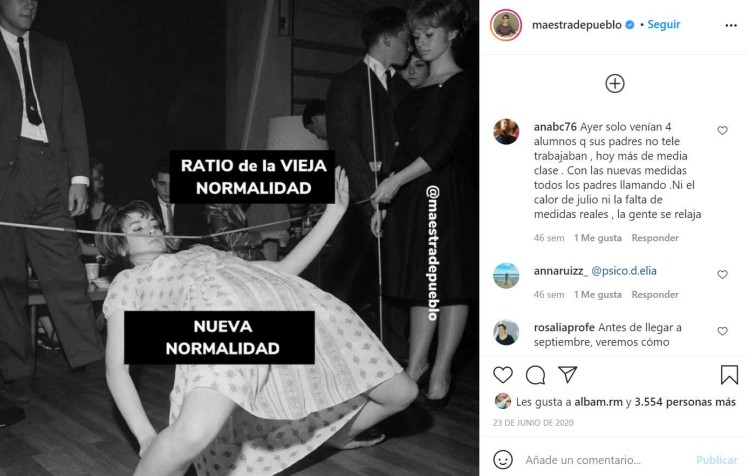

**Figure 11.** Example of a humorous publication. In this case @maestradepueblo makes a joke about the situation pre-Covid and post-Covid in the educational system. Source: @maestradepueblo.

Taking into account the mercantilist drift affecting Instagram, it is a priority to determine to what extent the influencers studied generate income through their activity on the platform in question. The digital marketing software "Infuencity" allows access to various data on the accounts involved in our study. The total earnings generated by influencers on the Instagram platform are presented by applying a mathematic algorithm developed by said software.

Table 7 shows the average earnings that the educational influencers analyzed obtain through the publications made, which generate mentions, engagements, shares, recommendations, along with other interactions from third parties (other influencers). All of this generates a value of the channel's diffusion, giving rise to the figures shown in the following table. The total value obtained taking into account all the educational influencers studied is EUR 6460.98, with figures that vary greatly from one influencer to another. In this sense, @applesandabcs and @maestradepueblo are the two accounts that offer the best data, EUR 1493.04 and EUR 2205.33, respectively, while @desdemiaula (EUR 14.94) and @enticonfio (EUR 8.54) are the ones with the worst figures, being very insignificant amounts.

**Table 7.** Average earnings of influencers.

| Influencer | Average Earnings |
|---|---|
| @2profesenapuros | EUR 310.42 |
| @3ways2teach | EUR 44.96 |
| @applesandabcs | EUR 1493.04 |
| @auladeapoyo | EUR 59.76 |
| @clubpequeñoslectores | EUR 167.86 |
| @desdemiaula | EUR 14.94 |
| @educacioilestic | EUR 45.66 |
| @enticonfio | EUR 8.54 |
| @entrenubesespeciales | EUR 434.20 |
| @maestradepueblo | EUR 2205.33 |
| @maestrosaudicionyl | EUR 376.88 |
| @teachinghumor | EUR 608.98 |
| @thinksforkids | EUR 690.41 |
| Average total earnings | EUR 6460.98 |

These values obtained through Influencity have been calculated through a mathematical algorithm that identifies the quality of user interactions.

## 4. Discussion

Educational influencers are making great efforts to gain a foothold on Instagram, generating a transmedia production that can be interesting for capturing new audiences, including families and teachers as shown in a recent related study [42]. These communicators differentiate themselves from the rest by designing communication and interaction strategies that are typical of influencer marketing. Their profiles, incorporated in this strategy, offer an attractive first image to audiences, through carefully created profiles, clear hypertext descriptions of the account, hyperlinked navigation, etc. [43].

In terms of audiences, the age ranges that predominate among these new registered audiences are populations between 25 and 45 years old, digitalized generations for whom social networks and digital environments are an essential part of their lives, the so-called millennial generation [44] (+P1); this fully agrees with the intention, detailed previously, that digital marketing follows to enhance the influence on this generation due to its close relationship with these interaction platforms [29]. The prominence of women and their age is significant, as this is a social group that may be involved in teaching or caring for a child, generating interest in the type of content that is shared on these accounts with the aim of improving the quality of teaching inside and outside the classroom, showing greater involvement than male populations and coinciding with the conclusions reached in a recent study by Fernández-Freire, Rodríguez-Ruiz and Martínez-González [45].

The behavior of audiences, through their interactions on Instagram in the COVID-19 pandemic period, makes it possible to know some characteristics of the media impact

that influencers generate through their creations and publications. Most of the accounts analyzed belong to those categorized as microinfluencers, as they have audiences of less than 100,000 followers, covering very specific topics as a strategy to specialize in a very specific audience [46]. The data analyzed reveal that there may be a relationship between the greater number of followers of the channels studied and the number of creations or productions generated by the influencers, following the tendency described by Peters et al. [24]. However, this growth does not occur proportionally among the different influencers, which may be explained by a variable of great importance with respect to social networks, such as the quality of transmedia production, coinciding with the approaches presented by the eCommerce & Digital Transformation Observatory [47]. Following this line, we have seen how the two profiles with the highest number of followers are dedicated to sharing educational material resources as a way of doing business and having an economic return, not with a purely educational and innovative purpose [38]; meanwhile, in third position is a profile focused on sharing comic creations on different aspects of current affairs related to the field of education. From the content analysis, we conclude that most of the profiles studied are focused on sharing material resources rather than other topics [36], with a high proportion of educational influencers using their Instagram profile to generate commercial activities by selling their own services or products or promoting other people's products, taking advantage of the great impact that their publications generate among their audiences (−P2), establishing differences with the results offered by Izquierdo and Gallardo, where a profile of an educational influencer focused on altruistic help to his followers is presented [33].

It should be noted that the data obtained show an audience that interacts more with those accounts whose purpose is to generate transmedia productions related to humor, where the meme format stands out. The high levels of interaction in these profiles is closely related to studies that present memes as outstanding formats due to the high levels of participation they generate and their capacity for vitalization in social networks [48,49]. The most used interaction resource among the audiences analyzed is the Like, consolidating the trend offered in various studies consulted [50,51]. Following the contributions of the Oxford Social Media Dictionary [52] it should be noted that, although the use of the Likes is a relevant element for analyzing the loyalty of new audiences, comments and mentions are the most transcendental communicative elements for producing a higher level of engagement between influencers and followers and, as a consequence, greater economic income from their impact. In this line of research, only the @thinksforkids account shows good levels of engagement, due to its high proportion of comments/likes (+P3).

The transmedia creations and productions shared by the educational influencers in this research show that they are concerned with maintaining a carefully curated and personalized feed, with the intention of consolidating the brand among their followers [25]. This means that, in general, the creations and productions they share have characteristics and styles marked by specific patterns that are recognizable to their audiences. A recent study [53] determines the use of these procedures in globally relevant influencers from other sectors (fashion, sport, celebrities, etc.), and once again highlight the increasingly widespread figure of the educational influencer as an expert and consolidated character in social networks as related studies determine [54]. Although Instagram is known for the great power of visual content, all the accounts analyzed make use of hypertextual descriptions to provide their productions with more information, revealing the importance of generating a digital narrative around the image, as Domenech has already pointed out in previous research [55,56]. In these hypertexts, educational influencers generally include textual elements that help to increase the impact of publications and accounts (hashtags, mentions, tags, emoticons, links to external websites, etc.) [22], fulfilling the dual function of providing information and increasing the impact of transmedia production or Instagram profiles on audiences [57]. Likewise, the use of messages to encourage interaction with new audiences is a fairly widespread practice among the profiles analyzed, in line with recent studies in which similar procedures are observed by influencers to increase participation

among their followers [58,59]. Users who make use of social networks, and in particular Instagram, are exposed to a high level of advertising bombardment, both explicit, with advertisements, and implicit, through the publications made by many influencers who encourage consumption among audiences [60].

## 5. Conclusions

Instagram is a social network that has one of the most active communities today, where users can find content adapted to all kinds of interests and where influencers look for their niches to spread their productions and transmedia creations. Educational influencers are quite consolidated and adapted to Instagram's own mechanics, with which they generate greater influence among audiences. Educational influencers are characterized by being users who care about the first impression they make on the community, as well as having a consolidated audience led by the millennial generation aged between 25 and 45, where women with an interest in the subject of "motherhood" stand out. The quantity and quality of the content have a significant influence on the growth of the profiles studied, with the use of the like being the preferred option for audiences to interact with the publications. This study has shown how educational influencers are involved in maintaining a careful and striking aesthetic in their channels; it also shows that the use of text in the descriptions is a widespread practice in publications, with communication adapted to the language of the post-digital society and that they implement different formulas to increase participation, interaction and impact on the audience. Despite the fact that the Instagrammers analyzed focus their activity on aspects related to education, it has been observed that digital marketing is becoming increasingly widespread among this type of influencer [61].

Although this study focuses on a fairly limited sample of influencers, it is a starting point for broader analyses in view of the constant increase in this type of profile, both on Instagram and on other social networks with a high social impact such as Facebook, Twitter or TikTok. Educational influencers are becoming new familiar faces and it is important that through their online activity they manage to place education in a relevant position for society as a whole, involving it in the great challenges of the future.

## 6. Limitations and Future Research

This research provides data of special relevance on the study of the educational influencers, especially during the outbreak period. However, it has certain limitations. The main limitation of this article is the reduced number of Instagram accounts analyzed. This situation has served to achieve concrete results; however, a larger sample would have provided us with a broader vision of the phenomenon of educational influencers.

Secondly, despite the relevance of the selected temporary period, an analysis of the accounts based on a longer time period would provide us with a more complete mapping of the educational influencers' relevance. At this point, future research will be developed, expanding our sample to other accounts (taking into account different nationalities and languages) and including a longer period of time.

**Author Contributions:** Conceptualization, J.G.-Q. and E.V.d.L.; methodology, E.V.d.L.; software, E.V.d.L.; validation, J.G.-Q. and E.V.d.L.; formal analysis, E.V.d.L.; investigation, E.V.d.L.; resources, E.V.d.L.; data curation, E.V.d.L.; writing—original draft preparation, E.V.d.L.; writing—review and editing, E.V.d.L.; visualization, J.G.-Q.; supervision, J.G.-Q.; project administration, J.G.-Q. and E.V.d.L.; funding acquisition, J.G.-Q. All authors have read and agreed to the published version of the manuscript.

**Funding:** This research received no external funding.

**Data Availability Statement:** The data presented in this study are available on request from the corresponding author. The data are not publicly available due to privacy permissions.

**Conflicts of Interest:** The authors declare no conflict of interest.

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
