# Peer review of "Educational Influencers on Instagram: Analysis of Educational Channels, Audiences, and Economic Performance"

_publications, doi:10.3390/publications9040043_

Round 1
Reviewer 1 Report
I believe that this topic is very interesting and worth pursuing. The interest in social media influencers is growing by the day and the research area has been splitting into segments. This is one of those segments - educational influencers. The research is, however, at its early stages and needs serious improvements, both for becoming better informed and to devise a more suitable research design. Please find my comments below, which I hope will be of help.
Regarding the introduction and research problem/objectives:
The paper lacks a proper introduction providing an adequate framing for the paper and a sufficient overview of the background to the research questions. Currently, the introduction is divided into two subsections (1) Influencers and new audiences: communication and interaction on Instagram and (2) Educational influencers and transmedia production. These are appropriate for the literature review section but not so much to provide an initial reasoning that drives the reader into the topic. For this reason, it is also not possible to understand which is the research problem, which is not clearly articulated, with an appropriate rationale and justification of its importance.
The authors use the two aforementioned sections to establish a context and jump right into presenting the research aim (“…how educational influencers use Instagram to consolidate their channel with new audiences, influence the community and make transmedia projection of their productions and creations”) objectives and hypothesis. Therefore, several questions are left without answers when the reader reaches the methodology section:
- What problem, gap, challenge do authors seek to solve or improve with this research?
- Why is it important to study the main characteristics of the profiles of educational influencers? Which are “main characteristics” and who defined them as “main”? The authors or the literature?
- Why is it important to discover the media impact that educational influencers have on new audiences? How are the research hypothesis aligned with this goal?
- Why is it important to analyze the communication and interaction of educational influencers with new audiences through their transmedia production?
- What is the relevance of gender in the study of influencers activities? This was not referred to in the literature review (gender issues) and there is no framing allowing the reader to understand why it is so important (H1: The audiences where educational influencers generate the greatest media impact are mostly women.)
Please attend to these issues. I would suggest adding a proper introduction, before the two existing sections of the literature review. While doing that, please frame the paper properly, introduce, discuss and refer to the relevance of the research problem, making sure to attend to the above questions. Please ensure that the ideas and justifications are well articulated and explained.
Regarding the title
I do not believe that the title is a good description of the paper and appropriate in length. What challenges are influencers facing? “new audiences” lacks further clarification. What is meant by “new”?
Regarding the abstract
It should include a clear reference to the research problem.
Regarding the keywords
After reading the paper, I do not think that these clearly represent the core issues that are dealt with in the paper.
Regarding the literature review:
The literature review is contemporary, but it is not comprehensive. It does not entail a critical analysis which further expounds the research problem, as further detailed below.
Regarding the methodology:
The research design is not clearly described, with adequate justification for the choice of methods and a clear account of how evidence has been analysed. In general, it does not demonstrate that acceptable norms of good research practice have been upheld in the conduct of research.
- Please clarify why a qualitative research book is referred to in this sentence “In order to move forward in a rigorous and scientific manner [28],”
- Please feel highly encouraged to illustrate the “categories” mentioned in lines 152-162. The text alone does not allow for a clear perception of relations and interdependencies of the categories.
- In lines 152-153 it is stated “three categories are established on which to apply the various analysis techniques, favoring a better treatment of data”. This sentence lacks a full stop and mentions 3 categories – the following sentence only refers to 2 categories. Please specify what type of categories are these. Are these categories for content analysis? Or are these dimensions of analysis which will be analysed according to sets of key variables? Please specify “various analysis techniques”.
- Please specify what is meant by “as well as certain quantitative data obtained through an observational analysis of the accounts analyzed on the Instagram platform.” Which data?
- What is the timeframe for the data collected and analysed? How was the data selected (the posts and audience interactions)? How many posts were included in the quantitative and qualitative analysis? Does this timeframe has any significant meaning to the research or is it totally random?
- Please specify the criteria used for sample selection. It is mentioned that the sample had to be narrowed down and to very specific types of profiles, but nothing else is explained. It lacks transparency. How many educational influencers in total have the authors identified? How was this search made? Did it only include Spanish educational influencers (geographical constricted sample)? Which were the criteria for sample selection? Please include this criterion in Table 1.
- I find that the content analysis (a qualitative component of the research) lacks transparency in the paper. Which procedures were used to analyse and categorize the content? Was this done in a systematic manner or is it a result of unstructured random observation/scanning of the Instagram profiles, based on the “feel” of the researchers? If this is the case, it lacks scientific soundness. When the effective content analysis is applied, the content classification model is explained and justified according to the literature (this results in a model that can be built by the authors, with dimensions and categories that can be static or dynamic, i.e. deductive or inductive, or both). After this, the categories are saturated with evidence (new categories might be added as a result of induction). Such a categorization of content (either textual or visual) then allows to systematically communicate the prevalence of categories (in a quantitative fashion) which can be reinforced/clarified/deepened with verbatim evidence – texts and media – in a qualitative fashion. The authors do not refer to any of these procedures on the methodology. A categorization model built on the literature needs to be provided (a table is preferred)
Regarding the results:
- The introductory text of the results section should provide an overview of the method/sections/structure adopted by the authors to present their results. Please rewrite the text in lines 201-205.
- Please specify how the data about the followers were obtained and which are, exactly, the variables used to classify the followers as real or doubtful. Was this manually verified by the authors? Please be consistent with terminology (real, doubtful, “nice” followers)
- Please make sure to insert all captions in English, as they are currently in Spanish.
- If SPSS was used, a couple of tests could have been used to provide significance to the research findings. In fact, in section 3.1.3, the authors refer that “With regard to the Followers/Followings ratio, it can be said that there are no significant differences between the accounts with the most followers and those with the fewest followers,”. Several other statements with no statistical support are made.
- As mentioned above, given the inexistence of an effective content categorization model the type of results presented in sections 3.2.1 and 3.2.2 appear to be selected at the pleasure of the investigator, in a one-sided, single-person perspective that lacks systematization. This paper is very opaque. For instance, when the authors state “The analysis of the images produced or created reveals that 62% of the influencers analyzed have a clear intention to take care of the aesthetics of the images that form part of their feeds, taking care of the quality projected from digital marketing (Figure 5).”. Which variables were considered for measuring “good aesthetics”? How likely is that this analysis is a result of the personal taste of the researcher? What constitutes “bad aesthetics” in this research? The only aspect I can identify is ‘consistency’, based on “images with the same type of colour tones and filters, or using standardized frames”, and “clear backgrounds” is the only reference to aesthetics.
- Moreover, please make sure to use the correct marketing/communication terminology. In lines 354-355 the authors state “elements are used as differentiating aspects with which to generate an identity image of the channel itself”. ‘Identity’ and ‘image’ are actually two opposed concepts. The ‘identity’ is built from within – it is what the organization assumes it is, in its own words. The ‘image’ is built from the exterior and consists of the sum of representations that the audiences have regarding the organization. I suggest reading this book: https://uk.sagepub.com/en-gb/eur/corporate-communication/book268492 Brands try to build an image that is consistent with the identity they have defined. This is why the copy strategy is very important, because it tries to translate the identity into the external image.
- Regarding the results presented in lines 361-368, these refer to personal disclosure. This is actually a very relevant topic in the field of influencers. I find these results interesting and relevant. However, the paper does not have sufficient literature background to allow for a discussion of these results. At this point, there are two key aspects that could be improved in the literature review: gender issues and personal disclosure. Moreover, in Figure 7 not all images are evidence of personal disclosure – at least not in the way defined by the authors as “share personal images of the owner, selling his personal life, family members, work 363 environment, etc”.
- In section 3.2.3, the authors refer “When analyzing the purpose of the transmedia production that the influencers under study share on their accounts, it can be seen that in most cases they share ideas for material resources so that they can be used in learning contexts by their audiences”. How were the posts analysed? How many? According to which content analysis categories? What were the authors looking for when analysing posts? Was it only to detect if a commercial purpose is present? Influencer monetization is a well established area of research. Again, the results (or their intent) is relevant but the paper lacks a background on this issue, which would allow to discuss these results and feed back to the literature or improve current practice in the field. The current relevancy of studying influencer monetization is not whether it is or is not used. It is ‘how’ it is used – monetization strategies. This could greatly improve the work that is being conducted.
- The results concerning the average earnings presented in lines 445-458 need to be better explained. I understand that they consist of a replica taken from the platform Influencity, but in a research report the methods and variables are key, and it is important for researchers to know exactly what they are communicating. Please state how is the quality of the user interactions measured and transformed into earnings.
Regarding the discussion and conclusion:
It is highly uncommon to present new literature in the discussion section. This is a clear indicator of not having a sufficient literature review in the paper and/or not having conducted relevant “related research”. Therefore, although a summary of the main findings is discussed it is not sufficiently detailed, with a depth of insight that provides a firm foundation for making a contribution of knowledge. This is absolutely crucial for the consistency of the research presented and, again, for the formulation of the research problem. Most of the analysed and discussed issues are confirmed by the existing literature brought to the discussion. If this literature was considered at the initial stages of the research design, the authors could have contributed beyond existing knowledge, not just confirming it to suit the analysis they conducted based on a somewhat random choice of variables.
This paper covers the following aspects regarding 13 educational influencers: A) on the quantitative side, the verified accounts, gender, age and general interests of followers; general Instagram post and interaction metrics; B) on the qualitative side: the posts’ aesthetics, personal disclosure, use of hypertext, influencer monetization and use of humour. These aspects should be systematized in a coherent research model, aligned with the research questions and research hypothesis. Since the research is as its starting point, as the authors refer, it would be crucial to revise the research and analysis model, the research design and key variables, and to build a very informative review of conceptual and related research review.
I also believe that this work lacks a method to analyse the influencers’ content strategy (the themes/topics/issues). The authors have compared the posts aesthetics, personal disclosure, monetization and sense of humour. Some of these could be categories of themes of a social media content strategy that would shed light on the discourse, identity and positioning of influencers (what do influencers talk about?) and allow to benchmark it among influencers (profiling them), while also analysing the audiences’ response per theme/topic. This could produce a proposal for content strategy for educational influencers to adopt when aiming at improving their influence for a given audience.
The word “conclusions” appears twice in the last two section’s names. The authors should acknowledge the limitations of their work, as well as the implications of their research.
Author Response
The word document with the changes made according to the recommendations is attached below.
Changes are marked in red with a comment of each case.

Reviewer 2 Report
Dear author/s
I read your manuscript with great interest. The topic you are discussing is really interesting and topical! However, the manuscript needs significant/major or minor changes that can be significantly improved. Below you will find some major or minor points in the manuscript which needs clarification, refinement, reanalysis, rewrites or/and additional information and suggestions for what could be done to improve it.
Section 1 (introduction) is too long. However, everything that is written is very important and may need further development due to the issue. I would suggest you to separate the literature review from this section and make it as a new section (after section 1 as background or/and literature review) with a small/minor revision / update and with up-to-date international references. Also, from the section 1 (introduction) the aim or/and objectives of the study or/and hypotheses or/and research questions are absent or/and unclear, and which should be numbered and clearly written. To help you, I quote some questions (as list of points) so that it can be included in your introduction:
-What is the importance of making this research/contribution that it brings to the literature in the field?
-Why should readers be interested?
-What problem/ gap resolve/fill this research?
-To fill this gap (resolve this problem) what solution/intervention/benefits does this research bring? (in other words, how the proposed study will remedy this deficiency/gap/problem and provide a unique contribution to the literature).
-What is the research question which address to the purpose of the research?
Some of these are already included or/and included in another section (e.g., Materials and Methods) and should be moved.
To summarize, this section should be reviewed and updated.
Section 2 (Materials and Methods):
From this section are missing some points and information, e.g., the type of methodology, information about the methods you used, the pilot phase, the reliability, ethics issues, the consent protocol, measuring instrument, the sample selection criteria, etc.
Although some of them are mentioned, a better reorganization / presentation may need to be done, and of course with a relevant reference in the literature where needed. Also, due to the (new) methodological approach followed (where there is already relevant literature and it would be good to mention), some of the above issues are not needed and the reason should be mentioned.
Most importantly, the type of methodology you follow and the type of analysis should be mentioned. You need to be clear and unambiguous.
Moreover, it would be good if you could mention further demographics of your research sample.
In summary, this section needs some minor revision for a better presentation. Also, as I mentioned above, some information should be moved to the first section (introduction).
I should mention that section 3 (Results), section 4 (Discussion and conclusions) and section 5 (Conclusions) are complete and with a very good interpretation.
I would suggest you, if you agree, to change the title of some sections, such as, for example, section 3 from "Results" to "Results and Findings" and section 4 from "Discussion and conclusions" to "Discussion".
After the revision of your manuscript you may need to revise or re-edit the remaining sections as well as the abstract as a final check.
Kindly check that the numbering or title of the sections or/and subsections is correct, and kindly check for grammatical errors.
As a final comment, I recommend a careful or/and an in-depth revision of your manuscript to improve it.
Author Response

(The authors gave the same response as above.)

Reviewer 3 Report
Congratulations to the authors for a very good piece of research, but it would be appropriate to include some additional information:
- What criteria did they use to choose the sample of these 13 Instagramers?
- How many followers does each one have? Where are they from?
- When was the analysis done?
- What variables were analysed?
Author Response

(The authors gave the same response as above.)

Reviewer 4 Report
Dear Authors,
Thank you very much for submitting the paper to the Publications journal. I read it with great interest. In my opinion, your study refers to an essential and current topic which is educational influencers and social media platforms. The study focusing on the abovementioned topic could be a valuable contribution to academia and society (especially when considering the popularity of the social media platforms or the changes in the widely understood education branch). However, I think that you could improve the quality of your paper. I found some significant flaws there, mainly in the paper's structure, literature review and methodology.
- Let me start with the title: it is too long and this difficult to follow. It is also inaccurate because it does not inform the reader that you focus only on Instagram influencers. I would advise you to change the title by narrowing it and exposing the primary topic of your research.
- One of the most problematic issues in your study is chaotic, blurry narration, vaguely defined research area and incoherence of content. There are too many mixed threads, and their descriptions and analyses are not set in proper, understandable sequences.
- The paper's structure needs to be reconsidered because now it lacks all the required elements.
- Your study does not give a proper "Introduction part". The 1.1 and 1.2 subchapters are more of a literature review nature. A reader may perceive that the article begins in the middle, without any general, opening reflections. Could you consider implementing a brand new introductory part? You could refer to the social media branch (its condition, power, situation) and changes it brought to the different aspects of human life, including education. Here could use some examples of new, "platform-oriented" educational phenomena, including educational influencers (it would be good if you could hint here an operational definition) and their activity on Instagram. You could put this social media in the centre of the paper right from its beginning. The "Introduction" is the best place to explain why you have decided to analyze the chosen subject, what you are aiming at and how the paper is structured.
- I would also advise you to rewrite the theoretical part. We could assume that the current 1.1. and 1.2. parts are literature review, but I would say that it remain insufficient, chaotic and superficial. Could you consider rewriting it and, at the same time enriching it with other threads and references? We must remember that a literature review should identify the academic gaps in your research area and justify the need to write the research paper. It also should result in detailed RQ1.
- So my advice for the literature review would be:
- Instagram as a social platform (analysis of social media: definitions, functions, roles, audiences; a detailed description of IG platform – why it is "social", what are its functionalities, how it changed the social media environment, what is offered to the world – like influencers)
- Influencers (detailed definitions, functions, advantages and disadvantages, business issues, different types – including educational influencers)
- Educational influencers on Instagram.
- You did not define the transmedia term – it should be added because, in several places, you use this term in different meanings. You should also precisely define the term "new audiences."
- They are no RQs; the objectives and hypotheses do not result directly from the literature review (on was basis they were formulated)?
- This part is unclear "These objectives are the reason for the research that helps us define what we want to study and are related to the following hypotheses". It suggests that you did not know what you would like to analyze; thus, you did not have any research plan at the level of methodology (and after the literature review). It may also suggest that your empirical activities were conducted accidentally without a well–thought structure. Could you consider changing these sentence?
- In lines 152-154, once you mention about two research categories, once – three categories.
- There is no earlier justification for focusing mainly on "young women teachers" in your research (line 159).
- Could you add some details to Table 1 – the number of followers and the starting time of the profiles?
- I find the methodological part chaotic. Could you first present research goals, research questions, hypotheses there? Could you precisely inform what research methods, techniques and tools you used? Could you divide the description of the empirical studies into stages?
- In the results, there is a "nice followers" term. Could you explain it?
- Could you explain the results obtained in a more detailed way? You present the statistical analysis results but could have you wondered (or studied) where these differences shown in Figure 3 or 4 come from? What do they mean for the influencers branch, especially for the education subgenre? How they are related to the impact on their recipients. How would you define "impact" itself (because it is an ambiguous term)? I do not fully agree that the data on Fig. 3 and 4 stand for the impact – to my mind, they are just quantitative (demographic) characteristics of the audiences. Also, you wrote, "the age range with the worst data is from 13 years 265 old to younger (1.69%)" – why do you consider it the worst? We cannot be so sure about it if we do not know about the content and strategy of the selected profiles. Maybe there are not addressed to the youngsters so the youngsters are not within their observers (that would mean that profile are appropriately targeted and the data obtained are not the worst). I think that a detailed description of selected profiles' content (aim, educatory specialization, target group) is undoubtedly needed here.
- I have a concern about significant generalizations I found in your study (i.e. Fig. 5). You transfer the results of statistical surveys to all profiles, presenting some averages and backing this up with random examples (even Table 5 does not result from a thorough analysis). Meanwhile, to examine the strategies of educational profiles on IG, it would be necessary to look at the content of each profile individually. Considering the diversity of content, audience and IG functions, it would be worth examining the specific features of each profile separately. In this case, a statistical view gives a certain numerical overview but does not offer an in-depth, detailed analysis. Studying the actual impact of profiles is not enough to look at the numbers of likes, comments, and shares. For each profile, it would be necessary to analyze the content of these comments - then it would be possible to talk about whether the posts engaged the audience (and if so, when and how). In the context of profile management proficiency, it would also be helpful to study how each influencer used IG functionalities (how they operated stories, IGTV, #, emoticons, tags, etc.). Going further, when analyzing content, it would be helpful to look at which posts result from probable product placement (with whom they work with, what they promote and is it coherent with their profile, what response such posts evoke). In addition, if you have counted the demographic variables of the audience (age, gender), it would be interesting to correlate them with the content of the posts, including the marketing topic.
- You used an unjustified assumption, "obtaining some kind of benefit from those advertised". If you do not have a proof for that, it would be better to avoid such overinterpretations (additionally: how did you exactly count the earnings? Could you specify it?)
- The results do not fully support the conclusions. There are simplified and not bonded with RQs and objectives. Additionally, you did not include the limitations thread in this part.
Dear Authors, I think that your paper needs fundamental theoretical, methodological, and empirical revision at this stage. You have gathered plenty of statistical data, but the level of presented qualitative data (which are as important as the quantitative ones) is insufficient. As for now, the paper needs to be significantly improved (maybe you could focus on one topic, like economic issues of educational IG profiles), revised and resubmitted for further review.
Sincerely.
Author Response

(The authors gave the same response as above.)

Round 2
Reviewer 1 Report
Thank you for your efforts in improving the paper. I have carefully analysed them but, unfortunately, I do not believe that they are sufficient, nor the paper has been properly groomed. Please find additional comments below.
In the abstract:
[EVdL2]: The research problem is still not clear. The authors refer to “the *challenge*” of how educational influencers use Instagram”. What does this challenge entail? If there is no challenge, the authors could simply assume that their work is a descriptive study aiming at reporting how educational influencers use Instagram to communicate and interact with their audiences. Also, this sentence needs to be revised “(…) influence through communication and interaction with citizens and produce its trans- 15 media message.”
In the introduction:
The great majority of the questions raised in the previous review are left unanswered. I do not wish to become repetitive, but I would recommend rereading those guidelines and improving the introduction in such a way to provide adequate framing for the paper, and a sufficient *overview* of the background to the research questions/goals. The introduction should create a full link between the core/main ideas that are further developed in the literature review and create a path to the framing of the research goals. This has not been done.
[EVdL5]: The introduction lacks proper referencing regarding the key ideas presented. See, for instance, “the figure of influencers has established itself as one of the most important, thanks to its ability to set the trends of the moment, having a fundamental role in the consumption of a large part of the population”. This is not a novel idea. Please revise and support the ideas presented. Moreover, “economic implications” and “profits” are not the same thing. Please clarify this issue. Additionally, the authors state “An exploratory study has been applied with which to better understand the problem raised (…)”. What is the problem raised? Up until line 44, no problem has been reported. The authors must either: a) identify a critical issue, challenge, difficulty faced by the educational influencers that they wish to solve, or b) refer to the general interest of reporting on the activity of the educational influencers and on how they communicate and engage with their audiences as a way to systematize knowledge that allows them to improve their social media strategies, either for consolidated influencers or for novices.
Then the authors proceed to mentioning “(…) through the contrast of hypotheses, using content and descriptive analysis and being the SPSS software if great utility. One of the main values of this research is the contribution to the scientific community on the figure of the educational influencers in our society; as well as clarify what type of activity they carry out and the relevance the reach on Instagram.” There are some problems here. Hypotheses are not “contrasted”, they are tested. In scientific research papers, please be exact. Such phrases are not suitable “and being the SPSS software if great utility”. The quality of the paper is not improved just because the authors state that they have used SPSS (particularly when no statistical tests are presented, at all). This is a very old and very wrong idea.
I would advise rephrasing the last two sentences of the introduction to specifically refer to the three research goals that are presented in lines169-172.
Once more, I must urge the authors to submit the paper to a native English language expert for a full revision, because the quality of the texts is very low.
In the literature review
[EVdL7]: The new sentences added in lines 89-92 required appropriate sources. There is much more to explain about the profile of digital natives and millennials
[EVdL10]: The definition of transmedia lacks appropriate sources.
In the materials and methods and results and findings:
There is no evidence that a statistical analysis was conducted over the data collected. For some reason, SPSS appears mentioned also in the introduction, which I don’t find suitable, and there are no tests to prove if the hypotheses were rejected or not. Just the short indication “(+H1)” – which test was applied? Looking at the data that was collected, the authors could have analysed at least the distribution of averages of age, gender and interests per influencer, for instance. Additionally, SPSS or any statistical procedures were not included in the “process” of the research (the research stages). This leads the reads to assume that it was not used at all. Please consider changing the hypotheses to propositions as the analysis presented is more of qualitative nature. This way it makes sense not to test the hypotheses.
I still cannot make sense of the structure of the categories and subcategories. The authors have introduced “new audiences” in line 201 and the sentence is not intelligible. In Table 1, the subcategory “News audiences” lacks its phrasing. What is studied?
In lines 246-247 the authors state that they have collected the data during the outbreak of the COVID-19 pandemic. This appears to be relevant contextual information, but it is not framed in the paper in any way. This is another crucial aspect. It is mentioned as such in the first sentence of the section Limitations and future research, but not in the abstract, nor introduction.
The sentence in lines 630-633 is not intelligible and has several typos.
In the lines 250-257 is where the authors should provide information about how the statistical analysis was conducted.
There is a typo in line 293.
There is no need to repeat the label “audience age” in Figure 4.
Overall remarks
The authors have mostly attended to operational changes in the paper but have neglected most of the more robust problems and recommendations. This has a great impact on the overall consistency of the document and on the perception of the quality of the research procedures. Both these aspects are not properly groomed in the paper, thus, the overall result is poor.
I would advise reviewing the previous recommendations very carefully and incorporating them or, in case the authors do not wish to incorporate them, to explain why they have chosen not to do so. In any case, a proper response to reviewers’ comments is good/common practice, as our main goal is to assist in improving the published material.
Author Response
We would like to thank the indications you are giving us to improve the article, they are helping us to advance in its improvement and to learn for future research. We hope that the changes made are relevant. Thank you again for your time.

Reviewer 2 Report
Dear author/s,
I re-read your manuscript with great interest again.
Congratulations for the effort made to improve the work.
However, there are still some issues that I noticed that could be improved.
Kindly re-read your manuscript again with a clear mind and make the necessary corrections.
Please add the corresponding references to the new pieces of your manuscript, otherwise they will be considered plagiarism. Additionally, you can re-check your entire manuscript and add references where needed.
The conceptual clarification of the concepts / terms used is absent or unclear from your manuscript (e.g., digital natives, millennials generation, social networks, influencers, etc.). Please add the interpretation to the relevant points or re-think what I mentioned in my previous review to create a new section as (theoretical) background or/and literature review.
Κindly check again for grammatical errors as a final check as well as the syntax throughout the manuscript. Where you write "..." please change them to "etc" or "so on" (e.g., lines 39, 112, etc.).
Also, please expand the discussion section a little further, and compare your results to the ones found in similar studies. In particular, please cite more of the journal papers published by MDPI where possible.
At this point, you should also review and update the references used in your manuscript. Almost all of your references are in Spanish, which is very difficult for potential readers to read or find for further study. New articles/researches have been published in the last 3 years, so it would be better to review the references or change or/and add a more recent references. To help you, I would suggest you take a look at the following links and use some of these references:
- https://www.mdpi.com/journal/education/special_issues/Technology-enhanced_Learning
- https://www.mdpi.com/journal/sustainability/special_issues/ict_sus
- https://www.mdpi.com/journal/sustainability/special_issues/cultural_heritage_storytelling_engagement_management_era_big_data_semantic_web
- https://www.mdpi.com/2673-5172/2/2
- https://www.igi-global.com/book/handbook-research-iot-digital-transformation/261128
- https://www.igi-global.com/book/advanced-methodologies-technologies-media-communications/208645
As a final comment, I recommend a more careful and an in-depth revision of your manuscript.
Author Response
Thank you very much for your indications and for the significant reviews you are making on our article. We hope you find all the improvements in a suitable way. Likewise, thank you for the links you suggested to find new references to reinforce our research.

Reviewer 4 Report
Dear Authors,
Thank you for sending the revised version of your paper. In many places, the quality of the text has changed positively. Still, at the same time, some of the corrections are unsatisfactory.
- The addition of the section on IG is superficial. There is a lack of precise information on which contexts/areas IG has been studied so far. It would also be helpful to show the advantages and disadvantages of IG, as this is an essential part of researching this platform.
- The definition of "digital audiences" is introduced inconsistently with the text's narrative and without reference to the source. Additionally, you describe "digital natives" and not "digital audiences" (after all, digital natives and digital audiences are not the same things). Also, I don't understand why you are focusing on Millennials if, e.g. generation Z is also proficient in social media/Internet. I'm afraid that narrowing the analysis to Millennials is not entirely correct. Besides, while referring to Millennials, you could also pay more attention to their characteristics in the context of educational preferences. There are numerous reports on this subject. Also, Millennials should be much more emphasized in other parts of the text, not only mentioned).
- Definition of "transmedia message": could you please indicate the source of this definition? It is far from the classic vision of, e.g. Jenkins (2006) ("Stories that unfold across multiple media platforms, with each medium making distinctive contributions to our understanding of the world, a more integrated approach to franchise development than models based on urtexts and ancillary products". Jenkins' successors also speak in a similar vein). Your paper does not analyze the different platforms that create a coherent vision of the personal brand of educational influencers. Nor is there a study of convergent content based on the flow and complementarity of content between platforms. What appears in the text is 1) an analysis of multimedia in IG posts; 2) elements of prosumerism (which fits into transmedia as a feature of the phenomenon, but not as an equal sign). Given this, I am concerned that your approach to the term 'transmedia' is controversial (which is why, in the first review, I asked for the definition).
- Section 1.2 still lacks a definition of influencers, their classification, their characteristics (positive and negative), the contexts in which the phenomenon is studied. The literature on the subject is wealthy, but it does not appear in the text (mainly, the findings from the first version of the paper are left without enriching them with new threads).
- H1 is not a strong hypothesis: it is hard to assume something without first doing an in-depth analysis of the phenomenon (here: Millennials, who, despite appearing in H, are not given a particularly substantive exposure in the literature review, results or conclusions).
- I stand by my opinion from the first review. I wrote: "Meanwhile, to examine the strategies of educational profiles on IG, it would be necessary to look at the content of each profile individually. Considering the diversity of content, audience and IG functions, it would be worth examining the specific features of each profile separately. In this case, a statistical view gives a certain numerical overview but does not offer an in-depth, detailed analysis. Studying the actual impact of profiles is not enough to look at the numbers of likes, comments, and shares. For each profile, it would be necessary to analyze the content of these comments - then it would be possible to talk about whether the posts engaged the audience (and if so, when and how). In the context of profile management proficiency, it would also be helpful to study how each influencer used IG functionalities (how they operated stories, IGTV, #, emoticons, tags, etc.). Going further, when analyzing content, it would be helpful to look at which posts result from probable product placement (with whom they work with, what they promote and is it coherent with their profile, what response such posts evoke). In addition, if you have counted the demographic variables of the audience (age, gender), it would be interesting to correlate them with the content of the posts, including the marketing topic".
Taking this into consideration, I can not, unfortunately, accept this paper for publication as I believe it needs further significant revisions.
Author Response
Thank you very much for the indications you give us about our article. We are working to produce a good document and your reviews are very helpful. We hope you find the changes made relevant. Thanks again.
